# Learning Dynamics and Control of a Stochastic System under Limited Sensing Capabilities

**DOI:** 10.3390/s22124491

**Published:** 2022-06-14

**Authors:** Mohammad Amin Zadenoori, Enrico Vicario

**Affiliations:** Department of Information Engineering, University of Florence, 50139 Firenze, Italy; enrico.vicario@unifi.it

**Keywords:** Partially Observable Markov Decision Processes (POMDP), Input–Output Hidden Markov Model (IO-HMM), failure avoidance strategy, stochastic generative model, stochastic system modeling

## Abstract

The operation of a variety of natural or man-made systems subject to uncertainty is maintained within a range of safe behavior through run-time sensing of the system state and control actions selected according to some strategy. When the system is observed from an external perspective, the control strategy may not be known and it should rather be reconstructed by joint observation of the applied control actions and the corresponding evolution of the system state. This is largely hurdled by limitations in the sensing of the system state and different levels of noise. We address the problem of optimal selection of control actions for a stochastic system with unknown dynamics operating under a controller with unknown strategy, for which we can observe trajectories made of the sequence of control actions and noisy observations of the system state which are labeled by the exact value of some reward functions. To this end, we present an approach to train an Input–Output Hidden Markov Model (IO-HMM) as the generative stochastic model that describes the state dynamics of a POMDP by the application of a novel optimization objective adopted from the literate. The learning task is hurdled by two restrictions: the only available sensed data are the limited number of trajectories of applied actions, noisy observations of the system state, and system state; and, the high failure costs prevent interaction with the online environment, preventing exploratory testing. Traditionally, stochastic generative models have been used to learn the underlying system dynamics and select appropriate actions in the defined task. However, current state of the art techniques, in which the state dynamics of the POMDP is first learned and then strategies are optimized over it, frequently fail because the model that best fits the data may not be well suited for controlling. By using the aforementioned optimization objective, we try to to tackle the problems related to model mis-specification. The proposed methodology is illustrated in a scenario of failure avoidance for a multi component system. The quality of the decision making is evaluated by using the collected reward on the test data and compared against the previous literature usual approach.

## 1. Introduction

### 1.1. Motivations

In the past few years, automatic controlling of stochastic systems [1] has been increasingly applied to the industrial world. However, numerous systems are still controlled by human experts such as medical practitioners, pilots, train drivers. In [2], a thorough study has been done to automate the personalized dosage prescription for the patients with blood clotting problem. Considering the aforementioned cases, designing an automatic controller by application of the collected trajectories, is a challenging task.

A viable way for formulating the problems is to formulate them as Partially Observable Markov Decision Processes (POMDPs), a usual approach for decision-making under uncertainty [3]. In the most recent research in this particular field, where POMDPs are applied, it is supposed that parameters are known a priori [4,5,6].

The collected trajectories from the interaction between the controller and the system consist of system observations, actions selected by the controller, and the collected rewards. Taking this notice into account that we do not know the exact state dynamics of the system and it is demanding to formulate the problem of optimal decision making on every possible system observation, we formulate the problem as a data-driven Partially Observable Markov Decision Process (POMDP).

To this end, we need to train a stochastic generative model to learn the underlying system dynamics from the collected trajectories afterwards finding a policy for optimal decision making on the trained model.

The general purpose of this research can be explained as providing an automatic controller (regarding an optimal policy) to closely follow the decision making strategy of the system expert in every possible noisy observation of the system state besides controlling the system in an optimal manner under particular predefined objectives and constraints.

In particular in this research, we tackle the problem of multi-variable optimization where we deal with optimizing the learned model over two distinct objectives:
Quality of learning the system dynamics from the collected trajectories.Optimality measure of decisions made by the policy solved on the POMDP defined on top of the trained generative stochastic model. In this research, the optmilaity of the decision making is measured by the collected rewards.

In this research the decision making task that we address is optimal action selection in the multi-component systems that the selected actions modify the system dynamics and the main purpose of the task is decreasing the number of system failures.

### 1.2. Usual Reinforcement Learning Approaches

Model-based Reinforcement learning (RL) is a well known field of research related to the decision making problem formulations, and even early works such as [7] investigated the learning models that are still useful for Reinforcement learning (RL).

Howeverm more widely, usual optimization methods closely incorporate a downstream decision-making task during model training, and are growing in popularity across machine learning, from graphical models [8] to sub-modular optimization recent decision-aware optimization efforts have explored partially-observed problems in both model based and model-free.

### 1.3. Separation from the Usual Reinforcement Learning Methodology

The point that separates the problem formulation that we consider from the studied methodologies in the literature is that we are interested in the cases that just limited number of data collected by the interaction between the controller (an human expert or the current controller of the system) and the environment are available. Besides considering this important fact that the data are composed of collected trajectories from different number of sensors that is prone to randomly missed observations.

A typical model-based Reinforcement Learning (RL) approach, such as [9], provide a variety of strategies for learning optimum policies based on predefined objectives through online interactions, and the model is handled as an abstraction, removing the requirement for an explicit model.

Furthermore, these methods are frequently used in an online learning environment, and they need the agent to explore all possible state-action combinations, which is not feasible in this particular case that we study due to the system failure penalty.

In this work, considering the fact that the parameters of the POMDP except for the number of possible states and actions are not known a priori, we need to learn the topology of the underlying model from the noisy observations of the system state vectors that are labeled by deterministic labels which can be considered as rewards. These labels are result of system execution that are probabilistic.

### 1.4. Limitations

In this research, we face three limitations in the introduced setting:The parameters of the POMDP are not known a priori so we need to learn them from the collected trajectories.There is no guaranteed way to examine the solved policy and evaluate the policy value on an online environment regarding the failure cost.Training a stochastic generative model by using data contains numerous missing observations

### 1.5. Contributions

In this research, to avoid the aforementioned model misspecification related problems, a novel optimization objective is adopted from [10] to train an IO-HMM [11], as the generative part that describes the state dynamics of a POMDP. Planning the POMDP using a Point Based Value iteration (PBVI) algorithm, the target policy $\left(\pi_{\theta }\right)$ will be found that can be evaluated to collect high rewards in the *test trajectories*. Since interacting with the online environment is not feasible, the policy value is calculated utilising the Offline Policy Evaluation (OPE).

As a specified case, we are interested in the study the applicability of the proposed methodology in the field of optimal controlling of systems composed of different number of components. In such systems, under-performance or failure of even one of the components can fail the whole system. Occasionally, it is likely that the components experience low performance or failure for some time steps.

We assume that it is costly and not possible to have a system expert make decision on selecting actions for each noisy observed situation and, in addition, in practice, it is hurdled to have a hand-made policy that have solution for all possible partially observations of the components’ levels. Moreover, since the observed patterns of components are different than each other, it is probable to miss some observations for some of the components irregularly.

To reduce the number of failures in such systems, often some *automatic policies* or controllers are used to choose appropriate actions based on the system observations [12]. To answer the aforementioned research questions, we clarify our novel contributions as:A novel problem formulation in the field of model learning and planning of stochastic systems which is specified in multi-component system failure avoidance strategy is discussed.Offline reinforcement learning and offline policy evaluation (OPE) are applied to avoid directly applying the policy to the online environment.An innovative synthetic environment is discussedSolving the problem of missing observations during optimization of the model parameters

### 1.6. Synthetic Environment

The applicability of the proposed approach is illustrated using synthetic yet realistic environment under discussed conditions which is generated according to the failure avoidance scenario in a stochastic system consists different number of components.

This paper is structured as follows. We start by describing the background foundations that our work is based on in Section 2. Section 3 the missing data mechanisms is discussed. Afterwards, Section 4 describes the problem formulation and Section 5 introduce the parameter learning and optimization objectives in Section 6 and Section 7 the experiment setup and results are shown and discussed.

## 2. Literature Review

POMDPs are used for decision-making on a vastly diverse range of applications. In [13] the navigation problem is mathematically formulated as a partially observable Markov decision process (POMDP), and then a motion policy is discussed once a POMDP is solved. In [14] The intrinsic trade-off between sensor power consumption and the risk of misclassifying a patient’s health status is investigated, and the problem is categorized as a POMDP. In [15], to construct an attacker architecture capable of challenging numerous WSN protection methods, deep reinforcement learning technologies are deployed.

In [16], the navigation issue is modeled as a Partially-Observable Markov Decision Process (POMDP), to generate a policy that translates a series of motion instructions to belief states and observations.

In the past few years, POMDPs are getting more popular in the domain of maintenance. In [17], POMDPs are applied to decision-making for highway pavement. In [18,19], POMDP models were formulated to generate a policy for bridge inspection. In [20], for wind turbines with finite horizons, seasonal-dependent situation-based maintenance strategies are produced.

In [21], POMDPs were used to perform a complete literature assessment in the field of inspection scheduling and maintenance planning. Although many maintenance optimization approaches are presented in the literature that are used to model sequential decision problems, learning POMDP dynamics by log data is not studied in detail.

In the studies in the literature of POMDPS, *IO-HMM* is applied as the generative model to learn the state dynamics of a POMDP such as [22,23,24], the common methodology is maximizing the data likelihood without considering the collected reward during the training phase, then, separately solving the POMDP by using a planning algorithm, we take this approach as the *baseline* in this research.

To the best of our knowledge, the closely related problem to our defined problem is studied in [12] with this difference that in the current research we assume all actions have identical costs and the main purpose of this study is to find a policy which can avoid system failures.

## 3. Problem Formulation

We consider a discrete-time finite-state stochastic System operating under a controller that repeatedly applies inputs from a finite set of Actions according to some unknown *behavior* policy. The system state takes values in a finite space and evolves over discrete time depending on input actions applied by the controller. While input actions are observed exactly at each step, the System state is obfuscated by a continuous-valued noise, and it can be observed only at a subset of time points determined by the controller policy.

The stochastic system we referring to in this research, is Markovian that state transitions are dependant to the last event besides the last input that are considered as the sufficient statistics. In this system, Markov condition is always preserved and system reach regenerates after each event. However, it is possible to have observations from the states of the system but this observations are noisy that make the system partially observable. There is no absorbing failure state in the system; however, the failed condition is defined on the simulations which labels them by deterministic labels as failed or working.

Collected observations do not cover the entire dynamics of the underlying system, and they are rather determined by the objectives of control, which is assumed to be applied in vivo to maintain the system within a safety-critical operation range.

In this scenario, we want to find a *target* controller policy able to provide feedback also when the system is out of the range of observed behaviours. This involves the following two problems:
We want to learn a model of the state dynamics of the underlying system, i.e., a model of how the system reacts to control sequences and how this appears in observation. In doing so, we cannot apply inputs just for the aim of observation, but we are instead constrained to observe only those inputs that were applied according to the unknown policy. In the particular problem formulation we consider, we have a determined data set, which cannot be extended to better explore the dynamics.We want to learn also a policy that can closely fit the actions that the unknown controller takes on the basis of the noisy observations, which to some extent comprises an instance of the apprentice learning paradigm.

The proposed abstraction and problem formulation may fit various application scenarios; one example in the failure avoidance of the multi- competent system is briefly outlined to exemplify the concept:

Multi component stochastic systems, for example, are a comparable example in industry. The controller’s primary purpose in these systems is to keep the system operating at safe levels in order to prevent costly repairs. To demonstrate the application of the described technique in the aforementioned area of research, we created a synthetic but realistic environment that nearly matched the characteristics of the stochastic system we wanted to investigate. It must be remembered that the underlying system’s state vector specifies the system state, which can only be observed through noisy sensors.

## 4. Preliminaries

### 4.1. Markov Decision Process (MDP)

We consider a discrete-time nonlinear stochastic system modeled by the Markov decision process (MDP).

A MDP is defined by the tuple (S,A,C,P,ρ), where $S\in R^{N}$ is the set of states,

*A* is the set of actions,

$c\left(s_{t},s_{t-1},a_{t-1}\right):S_{t}\to R_{t}$ is the cost function which maps every state at time *t* to a reward,

$P(s_{0}|s,a)$ is the transition probability function, and ρ(s) is the starting state distribution.

The state dynamics of an MDP is depicted in Figure 1.

*Policy*: A *policy* of an MDP (which also called a strategy or scheduler in the literature) is a way to choose an action in each state.

### 4.2. Partially Observable Markov Decision Process (POMDP)

Formally, a POMDP is a 7-tuple (S,A,T,R,Ω,O,γ), where *S* is a set of states, *A* is a set of actions, *T* is a set of conditional transition probabilities between states, R:S×A→R is the reward function, Ω is a set of observations, *O* is a set of conditional observation probabilities, and γ∈[0,1] is the discount factor.state dynamics of a POMDP is illustrated in Figure 2. Parameter set of a POMDP can be separated into two distinct parts that describe the state dynamics (S,A,T,O,Ω) and planing objectives (R,γ).

*Agent*: An agent can be a controller of a system or also can be considered as a system expert that we desire to model its behavior.

*Discount factor*: The discount factor γ influences how much immediate gratification is preferable to longer-term gratification. When γ=0, the agent is solely concerned with maximizing the expected immediate reward, but when γ=1, the agent is concerned with maximizing the expected total of future rewards. *Belief*: In the case of a POMDP *M*, belief B(M) is a probability distribution over *M*’s state space. Although we do not know the present state of the system, we may use the probabilistic dynamics of *M* to estimate the chances of being in each state of the underlying MDP.

A Markovian belief state allows a POMDP to be expressed as a Markov decision process (MDP) in which each belief represents a state.

The conviction MDP that results will be defined on a continuous state space. There exist infinite belief states *B* because there are an unlimited number of probability distributions over the states *S*, even though the POMDP has a finite number of states.

We use a famous problem given in [25] as an example of a POMDP. The following is a description of the tiger problem: A tiger is placed behind one of the two doors with equal chance, while the treasure is placed behind the other. You’re standing in front of two locked doors and must choose between them. You will be wounded if you open the door with the tiger (negative reward). However, if you unlock the treasure door, you will be rewarded positively. You have the option of waiting and listening for tiger noises instead of opening a door right away. In Figure 3 the methodology to solve this problem is discussed.

### 4.3. When POMDP Is Preferred over the MDP

We need to identify three key elements, all of which must be relevant to make a problem suitable to be modelled as a POMDP:
The system’s states can be specified, but no one choice (action) matches all of them. For vulnerable species, for example, estimations of upper and lower population abundance under various environmental circumstances and threats may serve as a natural characterization of system states. In this system, we would expect the best action to be state-dependent, for example, near-extinction population sizes to require reintroduction actions, low–moderate population sizes to require threat management actions and close to carrying capacity populations to require no action.The system dynamics are stochastic due to natural variability and/or control uncertainty. These dynamics are considered to be Markovian, which means that the probability of transitioning to a state at time *t* + 1 is solely dependent on the system’s state and action at time *t*. We predict, for example, that population dynamics are stochastic processes, and that the efficacy of management interventions such as reintroduction will be unclear. The state transition dynamics obey the Markov property given an acceptable time step and relevant information captured in the definition of state.Managers are unable to see the condition of the system accurately (state uncertainty). In contrast to the perfect observable example, the optimum option to implement an imperfect observation of the system state is complex and includes extra considerations such as the history of previous observations and actions. The issue is transformed from an MDP to a more difficult POMDP when state uncertainty exists.

Together, steps (1) and (2) define an MDP, while step (3) makes the problem partially observable [26].

### 4.4. Input–Output Hidden Markov Model (IO-HMM)

We consider an Input–Output Hidden Markov Model (IO-HMM) [27] as a framework for representing environments consisting of hidden states, inputs (actions that may affect the states), and outputs (observations from the states). Formally, an IO-HMM is defined as a tuple $<S,A,Z,T,O,b_{0}>$, where *S* is the set of states, *A* is the set of actions, *Z* is the set of observations, *T* is the state transition procedure in the way that $T(s,a,s^{\prime })$ denotes probability $P(s^{\prime }|s,a)$ of going to state $s^{\prime }$ by taking action *a* at state s, *O* is the observation function such that O(a,s,z) denotes probability P(z∣a,s) of perceiving observation *z* as a result of taking action *a* and arriving in state *s*, and $b_{0}$ is the vector of initial state distribution such that $b_{0}\left(s\right)$ denotes the probability of starting in state *s*.

Since the exact state of the IO-HMM is not known, we make a belief about the state. We represent the belief by a vector *b* where b(s) denotes the probability that the state is *s* at the current time step. The following update formula can be used to calculate the belief *b* at the next time step from the belief at the current time step, given the action *a* at the current time step *t* and the observation *z* at the next time step [28].

### 4.5. Optimal Action Selection Problem in an IO-HMM Formulated as a POMDP

As disused in [22] a partially observable Markov decision process (POMDP) is a problem formulation of an action selection problem for an IO-HMM. A POMDP is defined as a tuple *P* = $<S,A,Z,T,O,b_{0},R,Y>$ all the Parameters of Partially Observable Environments defined as in the IO-HMM, *R* is the reward function so that R(s,a) represents the immediate reward of taking action *a* in state *s*, and Y∈[0,1) is the discount factor which discussed in the background section. The main purpose of an agent is to maximize the expected discounted total reward by choosing an optimal policy.

In this introduced setting, there are two distinct optimization criteria which are observation likelihood and the collected reward by the solved policy on the POMDP by planning.

#### Describing the State Dynamics of a POMDP by Using IO-HMM

Since in this introduced setting we desire to model the dynamics of a system from the collected trajectories, we need to train a stochastic generative model for this purpose. In other words, a partially observable Markov decision process (POMDP) is a formulation of an action selection problem in an IO-HMM. Considering a POMDP with *K* discrete hidden states, *A* discrete actions, D-Dimensional observations, and deterministic rewards (e.g., a failed or working system). The generative model for states $s_{t}\in \left\{1,2,\ldots K\right\}$ and observations $o_{t}\in R^{D}$ across time-steps t∈{0,1,…,T} is defined by:(1)$$\begin{array}{c} p\left(s_{0}=k\right)\equiv \tau_{0k}:Initial\;state\;distribution,\\ p\left(s_{t+1}=k|s_{t}=j,a_{t}=a\right)\equiv \tau_{ajk}:States\;transition\;distribution,\\ p\left(o_{t+1,d}|s_{t+1}=k,a_{t}=a\right)\equiv N\left(\mu_{akd},\sigma_{akd}^{2}\right):Observation\;distribution \end{array}$$

The model that is described in Equation (Equation 1) is an *Input–Output Hidden Markov Model (IO-HMM)*, that actions are considered as inputs and observations are cosndiered as outputs.

As described above Model parameters are defined as θ:{τ,μ,σ,R} that τ describes the transition probability to the next state $s_{t+1}$, taking into consideration the current state $s_{t}$ and action $a_{t}$. modeling each observation $o_{t},d$ as a Gaussian distribution, with parameters $\mu_{d}$ and $\sigma_{d}$ considering the mean and variance when in state $s_{t}$ after taking action $a_{t-1}$. We decide to apply the independent Gaussian distribution across the *D* dimensions for simplicity.

The model parameters {τ,μ,σ} which maximize the likelihood of trajectories can be efficiently computed using the EM algorithm that is described in HMMs [29]. Finishing the POMDP specification is the deterministic reward function R(s,a), specifying the reward from taking action *a* in the state *s*.

### 4.6. Missing Data in the Collected Trajectories

In the current problem definition, based on a probability distribution p(ψ), there are missing data in some time steps in the collected observations due to the fact that it is not possible to always have all observations from all the sensors of all components in a system. Given this fact, we need to study the training IO-HMM with the presence of missing data for this end.

#### 4.6.1. Missing Observation Mechanisms

It is discussed in [30] that observations are missing in a model in different ways. In case the missing observations are dependant on past missed observations, this mechanism is considered as *non-ignorable missing mechanism*.

Otherwise, in case that the missing observation probability is dependant only on the past observations and the observation misses the distribution parameter ψ, this mechanism is called *ignorable* missed and it means:$$P\left(R|Y_{obs},Y_{mis},\psi \right)=P\left(R|Y_{obs},\psi \right)$$

Or in other cases when the missing probability is not dependant neither on the observed nor missed observations is called *missing completely at random (MCAR)*.
$$P\left(R|Y_{obs},Y_{mis},\psi \right)=P\left(\psi \right).$$

#### 4.6.2. Complete Data Likelihood and Learning from Missing Observations

We discuss complete data likelihood as a case when we consider all given and missed observations on the calculations. In [30], complete data are composed of observations *Y*, missing distribution *R*, joint distribution of *Y*, and *R* is factored such as
Pr(Y,R|ψ,θ)=f(Y|θ)f(R|Y,ψ).

Then the observed data likelihood over missing data is:$$L\left(\theta ,\psi |Y_{obs},R\right)\propto \int f\left(Y_{obs},Y_{missed}|\theta \right)f\left(R|Y_{obs},Y_{mis},\psi \right)dY_{mis}.$$
according to [31] inference of θ based on $L(\theta |Y_{obs})$ is the same as $L(\theta ,\psi |Y_{obs},R)$ when two criteria are preserved:
The model parameter θ and missing observation distribution parameter ψ are separable and the joint probabilities over these two-parameter spaces are their multiplication.Missing data mechanism is missing at random (MAR) that it means:
$$P\left(R|Y_{obs},Y_{mis},\psi \right)=P\left(R|Y_{obs},\psi \right)$$
or as a special case of missing completely at random (MCAR) that implies:
$$P\left(R|Y_{obs},Y_{mis},\psi \right)=P\left(R|\psi \right).$$
since the available observations are given periodically based on a prefixed cycle without dependency on missed or given observations, we suppose the missing mechanism is MCAR.

#### 4.6.3. Complete Data Likelihood

If *N* different subjects by different length of observations $T_{n}$ are considered, then the complete data log-likelihood can be written such as:$$l\left(\theta ;y,s,u\right)=\sum_{k=1}^{N}\left[ln\left(\pi_{S_{k,1}}\right)+\sum_{t=1}^{T_{k-1}}\ln\left(a_{S_{k,t}S_{k,t+1},a_{t}}\right)+\sum_{t=1}^{T_{K}}ln\left(b_{s_{k,t}y_{k,t}}\right)\right]$$

M-step can be written with usage of Backward-Forward algorithm. Thus, the only modification needed here is taking the switching parameters into account that in this research, they are different discrete inputs that consider as actions
$$P\left(S_{t+1}\right)=P\left(S_{t+1}|S_{t},a_{t}\right).$$

#### 4.6.4. Considering the Given Observations in Calculations

According to [32] to handle missed observations in EM calculations, the only parameters that are directly dependant on the observations are the elements of the observation matrix *B*, so the only needed modification is modifying the calculations to modify that matrix to take the missing observations into account. For the iteration *v* the calculations are such as:$${\hat{b}}_{ij}^{\left(v+1\right)}=\frac{\sum_{k=1}^{N}\sum_{t=1}^{T_{k}}\gamma_{k,t}^{\left(v\right)}\left(i\right)\times \delta \left(y_{k,t}=j\right)}{\sum_{k=1}^{N}\sum_{t=1}^{T_{k}}\gamma_{k,t}^{\left(v\right)}\left(i\right)\times \delta \left(y_{k,t}\neq .\right)}$$
that it means ignoring observations when they are not given and consider $b_{yt}=1$ and an indicator function $\delta \left(y_{k,t}=.\right)$ controls that if an observation is missed. All other calculations to train other parameters ($\pi_{0},A$) are exactly the same such as provided in [29]. Two auxiliary variables are introduced here, $\gamma_{k,t}^{\left(v\right)}\left(i\right)$ is the probability that the model is in state *i* at time step *t* whilst $\zeta_{k,t}\left(iju\right)$ is the probability of transition between states i,j at time-step *t* by the input *a* for the subject *k*.
$${\hat{\pi }}_{i}^{\left(v+1\right)}=\frac{\sum_{k=1}^{N}\gamma_{k,l}^{\left(v\right)}\left(i\right)}{\sum_{k=1}^{N}\sum_{i=1}^{c}\gamma_{k,l}^{\left(v\right)}\left(i\right)},i=1,2,\ldots ,c$$
$${\hat{a}}_{iju}^{\left(v+1\right)}=\frac{\sum_{k=1}^{N}\sum_{t=1}^{T_{k}-1}\zeta_{k,t}^{\left(v\right)}\left(i,j,a\right)}{\sum_{k=1}^{N}\sum_{t=1}^{T_{k}-1}\gamma_{k,t}^{\left(v\right)}\left(i\right)},i=1,2,\ldots ,c$$

#### 4.6.5. Model Parameters Optimization in the Presence of Missed Observations

In the case that the missing observations are considered missing completely at random (MCAR), there are two common approaches to train the parameters in training HMMs that can be applied in this work to train the IO-HMM which are discussed in [33].

As discussed on [33], one possible approach is to ignore the missing observations while another one is inferring the missed values from the given ones. In the introduced setting based on this assumption that the missing observations do not convey any useful information, we just ignore the missed values in the calculations.

#### 4.6.6. Planning the POMDP to Find the Optimal Policy

The upper envelope of a finite collection of linear functions of belief may be described arbitrarily tightly as the *value function* of a discrete-state POMDP [34]. Even with very tiny POMDPs, however, accurate value iteration remains intractable. Rather of conducting Bellman backups across all valid beliefs, we employ point-based value iteration (PBVI), an approximation approach that is substantially more efficient. Only a limited range of beliefs are backed up by PBVI.

#### 4.6.7. Policy of a POMDP in the Currently Introduced Setting

As described above, given a POMDP with parameters θ, it is possible to the belief $b_{t}$, a probability distributions over all states of the underlying MDP, defines the posterior over state $s_{t}$ given all past actions and observations (history):$$b_{t}^{k},p\left(s_{t}=k|o_{0:t},a_{0:t-1}\right):\;sufficient\;stastics$$

The belief state, can be computed efficiently via forward filtering [29]. The POMDP can be solved using a planning algorithm to learn a policy $\pi_{\theta }:R^{K}\mapsto R^{A}$, mapping any belief to a distribution over actions. The goal is to find a policy with maximum value: $V^{\pi }=\sum_{t=0}^{T}\gamma^{t}$, given the discount factor γ∈(0,1).

In Figure 4, it is shown that how the policy plays the part to decide the next actions in the currently introduced setting.

### 4.7. Definition of the Behavior Policy and the Target Policy

*Behavior policy $\left(\pi_{\beta }\right)$:* As an unknown policy that collects the trajectories by interacting with the online environment. This policy can be considered as the strategy of the field expert or the policy of the current controller that is not known.

*Target policy $\left(\pi_{\theta }\right)$:* The policy that found by planning data-driven POMDP.

### 4.8. Offline Reinforcement Learning

Offline reinforcement learning is a revived field of study that aims to learn behaviors using just recorded data, such as data from previous trials or human demonstrations, with no environmental touch. It has the potential to make significant progress in a variety of real-world decision-making problems when active data collection is either prohibitively expensive (e.g., in robotics, drug discovery, dialogue generation, and recommendation systems) or unsafe/dangerous (e.g., healthcare, autonomous driving, or education). This paradigm has the potential to remove one of the most significant barriers to moving reinforcement learning algorithms from the lab to the real world.

### 4.9. Off-Policy Evaluation (OPE)

There are settings regarding the damage cost of policies, it is not reasonable to evaluate a policy in an online environment.

A possible example can be given in systems with high cost maintenance that a policy is considered as a new controlling strategy based on the systems’ observations. Taking this fact into account that the failure cost is not tolerable, it can not be possible to evaluate policy values in real environments that are the working systems. Therefore, a mechanism is needed to examine the policy on the collected trajectories which are labeled by the deterministic labels.

However, by planning a POMDP, it is possible to have the target policy $\left(\pi_{\theta }\right)$, there is no guaranty that it works as expected in practice. Since we can not evaluate the policy on the *online environment* due to this fact that is costly, we need to utilise *Off Policy Evaluation (OPE)* [35], Which in this work, consistently weighted per decision, importance sampling (CWPDIS) [36] is utilized.

Let *D* denote a set of *N* trajectories collected under behaviour policy $\left(\pi_{\beta }\right)$, *r* the collected rewards, γ discount factor, and H is the history of past observations and actions. aforementioned methodology estimates the value of a policy $\pi_{\theta }$ as:$$V^{CWPDIS}\left(\pi_{\theta }\right)=\sum_{t=1}^{T}\gamma^{t}\frac{\sum_{n\in D}r_{nt}\rho_{nt}\left(\pi_{\theta }\right)}{\sum_{n\in D}\rho_{nt}\left(\pi_{\theta }\right)}:Policy\;value$$
$$\rho_{nt}\left(\pi_{\theta }\right)=\prod_{s=1}^{t}\frac{\pi_{\theta }\left(a_{ns}|H\right)}{\pi_{\beta }\left(a_{ns}|H\right)}$$

In the simulated environments, we suppose that the behavior policy $\left(\pi_{\beta }\right)$ is known and both $\pi_{\beta }$ and $\pi_{\theta }$ are stochastic. In the real data settings that the behavior policy is not known, it is possible to estimate the behavior policy via the k-nearest neighbors approach which is discussed in [37].

## 5. Solution Techniques

We model the System dynamics as an Input Output Hidden Markov Model (IO-HMM), that is trained using sensed data through application of an Expectation Maximization (EM) methodology based on gradients of the likelihood, ignoring missing observations, and using likelihood as a measure of model fitness.

We then formulate a problem of optimal decision for the System control. To this end, we consider a Partially Observable Markov Decision Process (POMDP) with state dynamics specified by the learned IOHMM, defining state rewards as a function of observations and setting the discount factor as an hyper-parameter that determines the trade-off between immediate and long-term rewards. We use Point Based Value Iteration (PBVI) to find the target policy that maximizes the collected rewards v(π), borrowing the Off Policy Evaluation (OPE) technique from the context of offline Reinforcement Learning (RL). In this way, we measure the collected reward of the *target* policy by comparing to the decisions made by the *behavior* policy and estimating the collected reward. To this end, a technique from the literature based on importance sampling is applied to solve the mismatch between behavior and target policies. Collected data are split into a train and a test set, and the collected reward is measured using the target policy without directly applying the selected actions.

The IOHMM’s likelihood and the reward collected by the target policy, both evaluated on the test data, are competing objectives in this optimization task.

We use Lagrange multipliers to govern the trade-off between IO-HMM probability and the collected rewards objective during parameter optimization since we must account both of these opposed goals.

The IO-HMM is trained using the POPCORN approach, and the POMDP is solved to discover the best policy with a competing optimization target.

The gathered reward goal is more essential than the trained model’s data likelihood in this optimization job. As a result, the IO-HMM model must not be over-fitted to the data probability, otherwise the target policy will be unable to collect larger rewards on the test data. This problem is referred to as model mis-specification.

The POPCORN [28] optimization objective can handle the problem of model mis-specifaction raised due to the noisy observation collected from the system state.

The usual approach to find optimal policy in the POMDPs based on the IOHMM that is know as “two stage solving” is considered as the baseline approach that we compare the experiment results with that. In this baseline approach, the IOHMM trained and then solved on the separated stage with this regard the during optimization of parameters the collected reward objective is not taken into consideration.

The proposed solution technique is depicted in Figure 5. As described in Figure 5 in the diagram the collected trajectories are separated to the train and test sets and then train set is fed to the training process of IO-HMM. The next step is solving the POMDP that is formulated based on the IO-HMM to find the Policy π that afterwards Offline policy evaluation (OPE) methodology is used to evaluate the collected rewards by using the found policy and the test set.

### 5.1. Parameter Learning and Optimization Objectives

#### 5.1.1. Parameter Optimization Objective

In the task we consider, we learn the model to have the maximum likelihood of producing the observed trajectories given that a minimum reward goal which is calculated using Offline policy evaluation (OPE) is always reached, that is, a problem of maximum optimization with a restriction on the reward function.
$$\max_{\theta }\;\;L_{gen(\theta )}\;subject\;to:\;V\left(\pi {}_{\theta }\right)>\epsilon$$
ϵ is the minimum acceptable policy value. This is the usual approach which is known as the two-stage approach, but as discussed, this methodology is prone to model misspecification.

In this research, we use the idea that is applied in [28] inspired by the methodology introduced [38] By using Lagrange Multiplier λ>0 the non constrained optimization criteria is:$$\max_{\theta }\;\;L_{gen(\theta )}+\lambda V\left(\pi_{\theta }\right)$$

By applying the aforementioned methodology, it is possible to compare with both of these baseline approaches, referring to the λ=0 case as “2-stage” which is considered as the baseline in this work, and the λ→∞ case as “Value-only”.

In this approach, there are two distinct optimization objectives:

Log marginal likelihood:

$L_{gen(\theta )}$ as the log marginal likelihood of observations, given the actions in D and parameters θ:$$L_{gen(\theta )}=\sum_{n\in D}\log p\left(o_{n,0:Tn}|a_{n,0:Tn-1},\theta \right)$$

Collected reward objective:

Computation of $V(\pi_{\theta })$ entails two distinct parts: solving for the policy $\pi_{\theta }$ given θ using Point based value iteration (PBVI), and then estimating the value of the policy using OPE using the collected data D.

The IO-HMM likelihood marginalizes over uncertainty about the hidden states, can be computed efficiently via dynamic programming which is explained in [39], and is also possible to use *automatic differentiation* with regards to the θ.

#### 5.1.2. Parameter Optimization Strategy

The strategy for optimizing the aforementioned objective is to use the *likelihood gradients*. The objective is optimized with Rprop [40] with default settings. Rprop is a well-known algorithm that was originally developed to train neural networks. In the current paper, Rprop is employed to find parameter values that increase the objective function. The modification of Rprop to train HMMs in case gradients are used can be found in [27].

By design, Rprop only needs the sign of the gradient of the objective function with respect to each parameter. We take varying numbers of restarts as assessed by training objective values before the final assessment since the goal is non-convex and even the generating term permits numerous local optima.

To optimize the parameters of the model we used different number of epochs. One epoch means that each sample in the training dataset has had an opportunity to update the internal model parameters. An epoch is comprised of one or more batches.

## 6. Experiments

To exemplify the concept of the proposed approach, we design empirical experiments aimed at answering the following questions:
**RQ1:** (The problem of miss-specified model)Does the proposed approach permit to control the trade-off between the likelihood of the learned IO-HMM and the rewards collected during the traning epochs in the synthetic environment by the optimal POMDP target policy?**RQ2:** (Studying the affect of the noise level on the collected rewards)How does the noise of the collected observations in the designed environment affect the policy collected reward?

It is noticeable that in the research question it is emphasised that the applicability of the proposed methodology is just confirmed in the synthetic environment under particular explained circumstances.

To this end, we address the application scenario of the multi-component system introduced in the problem formulation section, considering a component based stochastic system, whose state vector is observed through noisy sensors

Specifically, we consider a stochastic System made by a collection of concurrent components, by a finite set of states which actions modify the probabilities of the transitions between higher and lower levels, and it is assumed that the controller shall maintain the state vector within a certain safe working levels.

To this end, we design two experiments by using the simulated environment:

Experiment 1

In this designed experiments we want to show the collected reward objective comparing the POPCORN and the baseline approach.

Experiment 2

We want to compare the collected rewards by using the POPCORN methodology and baseline approach optimization objective by applying different level of noise to the collected observations.

### 6.1. Definition of the Simulated Environment

We consider a system composed of different *C* components and *L* possible exact levels for each component. These components are observed periodically and it is probable to have some missing observations occasionally for some components irregularly. The task we consider here is designing a controller for the purpose of failure avoidance in the multi component system consists finite set of discrete Actions.

In the aforementioned system, there are *M* different actions that modify the dynamics of the components stochastically. We define a safe performance range for each component, trying to keep all components in that range. An MDP is used for modeling the simulated environment and then by adding *Gaussian noise* we make the system partially observable.

In this way, the introduced environment can be similar to real-world systems in that each component is sensed by using a different number of sensors.

**States:** We consider there are $L_{c}$ different possible exact level for each component *c*. For each component $l_{c}\in \left\{1,2,\ldots ,L_{c}\right\}$ is the exact level of a particular component *c*. The exact state of the system is a vector of length *C*, number of components, so there are $\prod_{c=1}^{C}L_{c}$ possible states in this system and $s_{t}\in N^{C}$. One possible state vector is shown in Figure 6.

**Observations:** Observations $o\in R^{C}$ are the vector of noisy observations from the exact state of the system. As an example, if the exact state of the system is [2,2,3,2,1], noisy observation of the system state is a vector such as [1.34,1.89,3.24,1.78,0.87] for his particular case.

**Actions:** There are *M* different actions that affect the current system state and transition probabilities stochastically.

**Trajectories:** Trajectories are at most *T* time steps for each simulation.

**Rewards:** Rewards are sparse, with 0 reward at intermediate time steps and Failure cost of the failed component or +1 for the working trajectories at termination that is labeled based on the observations of the components. In the case that for *J* time step one or more components are out of the safe working range, we consider the system as failed and label the final state by −1.

**Data generating policy:** We suppose actions are selected by the behavior policy ($\pi_{\beta }$). However, the deterministic optimal policy $\left(\pi^{*}\right)$ of a MDP can be easily found by using the Value Iteration (VI) algorithm [41], we consider that the behavior policy ($\pi_{\beta }:p\left(a_{t}\right)$) in the underling MDP is a non-deterministic ϵ-greedy policy.

The policy selects non-optimal actions with probability ϵ that By using this policy, we try to show that the policy of the controller is not perfect and it is prone to make mistakes. We consider this aforementioned policy as the sub-optimal policy $\left(\pi^{\epsilon }\right)$.

**Data Generation:** 2500 trajectories are generated under an ϵ-greedy behavior policy, given observed trajectories, we train an IO-HMM and then solve the POMDP to evaluate the policy via an additional 2500 trajectories to the test data.

All experiments were performed on a 64-bit Intel I7 4690k @ 4 GHz CPU with 16 GB RAM on python PyPOMDP [42] An educational project with modules for creating a POMDP (Partially Observable Markov Decision Process) model, implementing and running POMDP solver algorithms, and for the matrix calculation NumPy [43] a library for the Python programming language, adding support for large, multi-dimensional arrays and matrices, along with a large collection of high-level mathematical functions to operate on these arrays are used.

### 6.2. Experiments Setup Specification

In this work, we introduce an empirical model to examine the failure avoidance strategies in a stochastic system. We consider 5 components and 8 actions which affect the dynamics of the components stochastically. Respectively, there are 3, 3, 2, 5, 2 ordinal possible levels for each component.

By considering these exact levels for each component, there are 180 possible exact system states for the whole system. The initial dynamics of the components are described in Figure 7.

The components dynamics are defined in a way that they reach the final state faster than going back to the lower working levels. The actions are considered to modify the pace of the system level transitions to make it slower or faster.

As previously discussed, the actual state of the system is completely known and may be detected solely through the use of noisy sensors that make the system partially sensible. The observations examined created by varied Gaussian noise levels from the underlying MDP, because larger noise levels in the observation mechanism make the training process and solving the POMDP more difficult.

We define a unique *safe working level* for each component that is described in Table 1. These components can be considered as different inter-dependent units that are desired to stay in the safe level.

The aforementioned components’ dynamics in the Figure 7 are modifiable by the actions that are described in the Table 2. We consider an action sequence consisting of 3 binary actions that can be activated at the same time. Therefore, there are eight possible different actions in this setting. The main idea behind designing these inputs are speeding up the return from the maximum levels to the lower levels.

It is assumed that the level of Component #5 affects the Action # 4 effects, which in this way we suppose that the components are inter-dependant. In this setting, we suppose that Component # 4 and Component # 5 are works independently from the action effects.

We consider J=3 that it means, if in the system for more than 3 time steps one or more components are out of the safe level, we label that state vector as failed by −1.

For the purpose of reproducibility of the experiment results, in Table 3 hyper-paramters during the generating the synthetic data are given.

To be a substantiation of a sensor system in the observations from the synthetic environment, there is an added Gaussian noise with level of σ to the true state of the system as
$$o_{t}=s_{t}+N\left(0,\sigma \right)$$
where the additional noise to the true state of the MDP makes the system partially observable.

### 6.3. Comparing the Collected Rewards during Training (Experiment 1)

In the first experiment, during training, the noise distribution and the number of epochs are pre-fixed. We want to measure the collected rewards of the target policy by adopted the methodology from the literature that are used the competing objectives. We consider the minimum change in the collected rewards during training of 50 epochs in one of the approaches that is happened in 300 epochs in the model training by using observations generated by random Gaussian noise with level of the σ=0.3. Every 50 epochs in the experiments, the collected policy values on the test data are plotted.

In this research, the collected reward is the measure to compare the different policies. Since the number of positive rewards in the collected trajectories in the collected data is not mandatory higher than the negative ones, it is probable that the collected reward on the test data be a negative value.

In the experiments, we need to compare the collected policy value, which can closely be interpreted as the number of failures in the test data by the baseline approach and proposed methodology. Here λ=0 is referring to the baseline that is known as the two stage POMPD solving approach while λ=3.16 is the hyper-optimized λ and is referring to the applied methodology which is the optimization of the competing objectives.

As the results of the experiments show, the policy found by the utilized methodology collects more rewards than the two-stage solving approach during the 300 epochs, while the baseline approach is just learning the model parameters that maximises the data likelihood objective. The results of the experiments are shown in the Figure 8.

### 6.4. Collected Policy Values under the Different Observation Noise Levels (Experiment 2)

In the second experiments, we want to answer the second research question that we asked in the first section, we utilize the data sets that consist of 2500 train and 2500 test trajectories which are generated by different Gaussian noises. In these experiments, 300 epochs are used and the *average collected rewards* on the test data measure the performance of the policies.

Synthetic noise may be added into data not just as new variables, but also as direct effects on the true values. The researchers can assess the sensitivity of the analysis results to random noise of the predictors by analyzing the data set with noise inserted in this way. The noise magnitude can be adjusted to assess how much noise the technique for evaluating the data can bear before its performance decrease.

Assume that when the unchanged data set was evaluated, a certain set of predictors had greater value than synthetic noise factors. Increasing the quantity of noise in the system estimates the point at which the relevance of these predictors and noise variables begin to consider. Varying predictors have different threshold noise levels at which their influence can no longer be observed.

The random noise has been used to make the identifying the system state more challenging that in the Table 4 the levels of the noise is depicted.

By comparing the policy values, it is possible to compare the performance of the baseline approach and the adopted methodology. The collected reward by the baseline approach is shown by $\pi_{\lambda =0}^{\theta }$ whilst $V_{\pi_{\lambda^{*}}^{\theta }}$ shows the collected reward by the novel adopted optimization objective.

In conclusion, based on the results of the different experiment settings considering the different noise levels (σ∈{0.3,0.4,0.5}), collected policy values ($V^{\pi_{\theta }}$) by the application of the novel optimization objective is higher than the baseline utilised approach.

### 6.5. Answering the Research Questions

By discussing the results of the designed experiments we answered the research questions asked in the Section 6 as below:Answering RQ1 (Tackle the problem of miss-specified model):Referring to the results of the first experiment, during the training epochs the collected reward by the application of the applied methodology is higher than the collected reward by the baseline approach.This important notice is that the training process by using the baseline model is over-fitted to the data likelihood of mis-specified model and could not collect the more possible rewards by the selected policy.This fact is based on the novel optimization objective used in the proposed approach that control by using the Lagrange multiplier (λ) that considers the collected reward as an optimization objective besides the data likelihood to tackle the model mis-specification problem.Answering RQ2 (Affect of the noise level on the collected rewards):Referring to the results of the second experiment the collected reward by using the applied methodology is higher in different levels of the random Gaussian noise that shows model mis-specification can be planned using the proposed methodology better than the baseline approach.This important fact shows that in the problems defined on this sensor system defined in the synthetic environment the proposed approach performs better to model the dynamics of the stochastic system by using collected trajectories.

## 7. Conclusions

We have presented an approach that can be illustrate by a problem definition to find failure avoidance policies through solving a POMDP by using the collected trajectories from an online environment. The applied methodology takes the log of the interaction of the existing controller with the system as input, then trains an IO-HMM as the generative model of POMDP state dynamics, to find a policy that can collect high rewards on the test data.

The applicability of the proposed methodology is evaluated on the noisy settings that based on the two designed experiments by measuring the collected rewards performance can be comparable the competing baseline. In this research, the problem of multi component system failure avoidance considering model mis-specification and different level of noise are solved that is illustrate using a synthetic environment.

The proposed solution has the benefit of not requiring the present controller to actively investigate the repercussions of actions in all scenarios, which might result in prohibitive failure costs in the real system. By deigning experiments it is shown how the applied methodology compete against the usual literature baseline approach to collected more reward by the noisy settings in controlling multi-competent system for the purpose of the failure avoidance. The collected rewards in different settings show the confirmation of the performance of proposed methodology.

In future work, by considering different costs and execution times for each action, it would be possible to consider the applied action sequence cost as another optimization objective. Our next goal is to investigate a stochastic distributed system by using the Markov Decision Petri Nets [44] in ORIS [45] as the underlying stochastic system to design a more practical synthetic environment. 

## Figures and Tables

**Figure 1 sensors-22-04491-f001:**
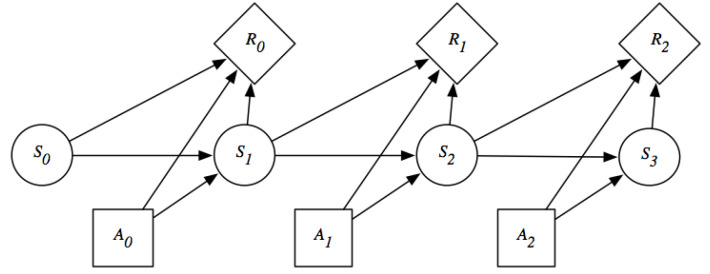
Probabilistic graphical model of a MDP. $S_{t}$ is the state at time *t* and $A_{t}$ is the action at time *t* and $R_{t}$ is the reward at time *t* that is dependent on the state of the system at time *t* and time t−1 and action taken in time t−1.

**Figure 2 sensors-22-04491-f002:**
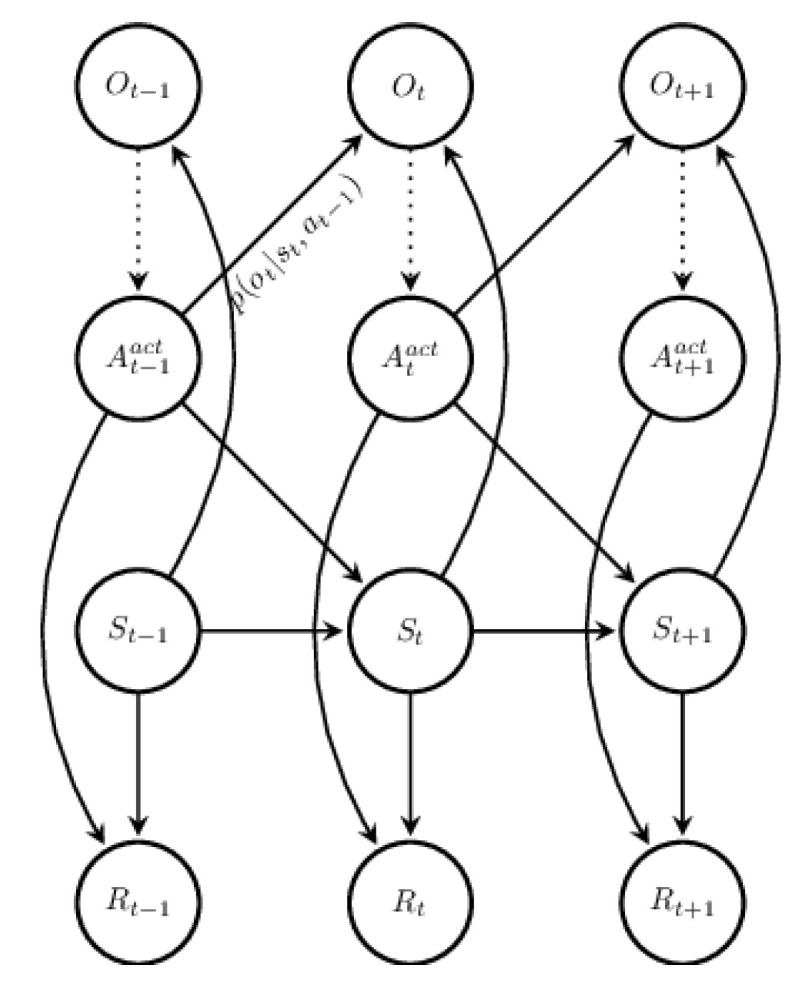
Probabilistic graphical model of the state dynamics of a POMDP. $O_{t}$ is the observation at time *t*, $A_{t}$ is the action at time *t*, $S_{t}$ is the system state at time *t*, and $R_{t}$ is the reward at time *t*.

**Figure 3 sensors-22-04491-f003:**
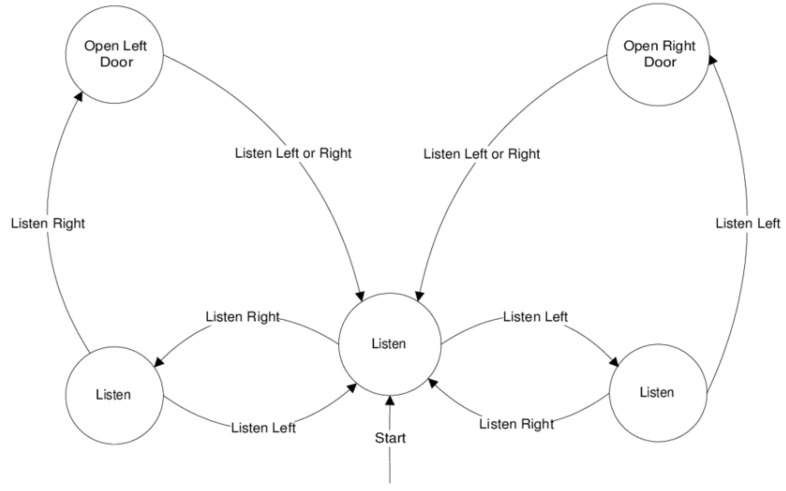
Policy graph of the tiger planning problem. Circles of the current state of the defined POMDP, arrows show the actions, and names on top of the arrows are name of actions.

**Figure 4 sensors-22-04491-f004:**
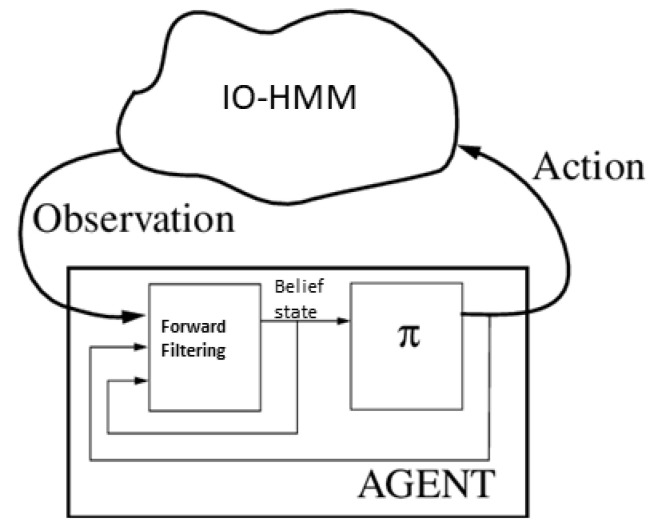
Different parts of the POMDP agent in the current introduced settings. π is the policy of the agent which based on the belief state that is computed using forward-filtering selects the next action.

**Figure 5 sensors-22-04491-f005:**
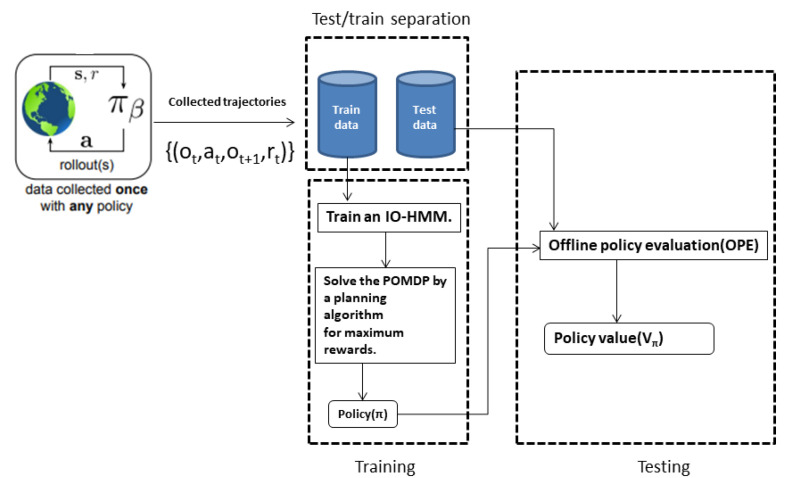
Learning and evaluating the target policy $\left(\pi_{\theta }\right)$ by the collected trajectories from the online environment. The behaviour policy ($\pi_{\beta }$) can be the controlling strategy of a system expert. $a_{t}$ is the action at time *t*, $o_{t}$ is the observation at time *t*, and $r_{t}$ is the reward at time *t*. The Collected policy value of the target policy measures the quality of decision making.

**Figure 6 sensors-22-04491-f006:**
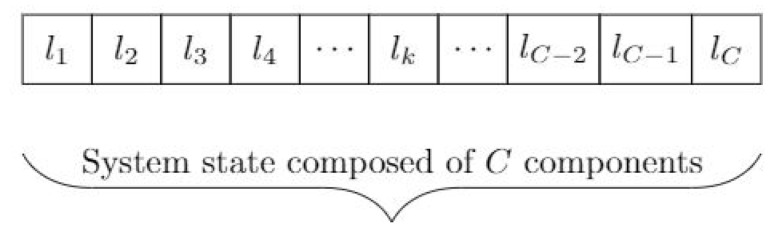
One example of the possible system exact state vectors. Each part of the vector shows the exact state of each component in the system.

**Figure 7 sensors-22-04491-f007:**
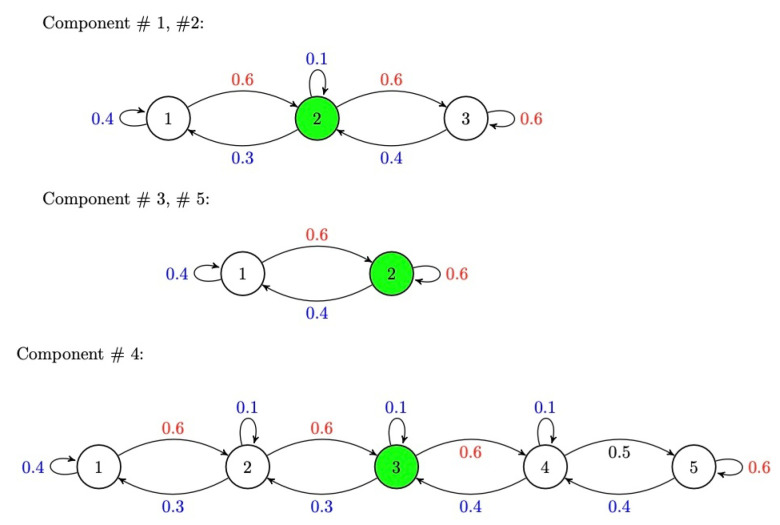
Initial dynamics of each component that these dynamics are modifiable by the actions. The shown states are the possible working levels for each component. As shown above, the components tend to transfer to the higher levels with higher probabilities. The safe working level for each component is marked green.

**Figure 8 sensors-22-04491-f008:**
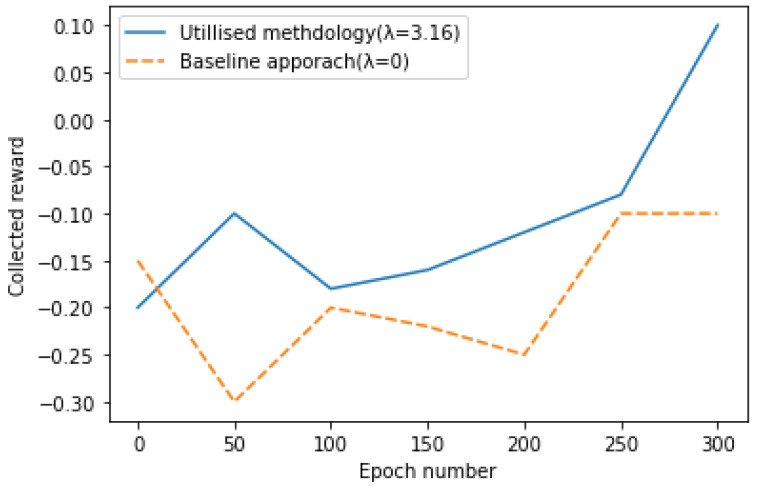
Policies values on the test data by applying the baseline approach and the proposed methodology.

**Table 1 sensors-22-04491-t001:** Components’ safe working range description.

Components	Safe Level
Component #1	level 2
Component #2	level 2
Component #3	level 2
Component #4	level 3
Component #5	level 2

**Table 2 sensors-22-04491-t002:** The effects of actions on the components dynamics. After taking the actions the dynamic of the components will be changed.

Actions	Effected Components
Action #1 turned on	Component #1, Component #2: level 3 -> level 2 with probability 0.8
Action #1 turned off	Component #1, Component #2: level 3 -> level 2 with probability 0.4
Action #2 turned on	Component #3: level 1 -> level 2 with probability 0.8
Action #2 turned off	Component #3: level 1 -> level 2 with probability 0.6
Action #3 turned on	If Component #5 is in level 1: Component #1: level 1 -> level 2, with probability 0.3 level 2 -> level 3 with probability 0.7 If Component #5 is in level 2: Component #2 : level 2 -> level 3 with probability 0.9, level 1 -> level 2 with probability 0.5, level 1 -> level 3 with probability 0.4
Action #3 turned off	If Component #5 is in level 1: Components # 1 : level 2 -> level 1, level 3 -> level 2 with probability 0.1 If Component #5 is in level 2: Component # 2: drops 1 level with probability 0.2

**Table 3 sensors-22-04491-t003:** Hyper-parameters during generating the synthetic data.

Hyper-Parameter	Value
Number of Epochs	300
Discount factor (γ)	0.7
Maximum time of simulations (T)	15
Optimized Lambda (λ)	3.16
Number of components	5
Number of train data	2500
Number of test data	2500
ϵ-greedy policy	0.21

**Table 4 sensors-22-04491-t004:** Collected rewards by Target policy $\left(\pi_{\theta }\right)$ under the different observation noise levels (σ).

Noise Level (σ)	$V_{\pi_{\lambda =0}^{\theta }}$	$V_{\pi_{\lambda^{*}}^{\theta }}$
0.3	−0.70	−0.49
0.4	−0.64	−0.53
0.5	−0.82	−0.60

## Data Availability

Data is available in request.

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
