# Peer review of "Learning Dynamics and Control of a Stochastic System under Limited Sensing Capabilities"

_sensors, 2022, doi:10.3390/s22124491_

Round 1

Reviewer 1 Report

The article is devoted to the problem of learning dynamic systems of a stochastic nature in conditions of limited perception. Of particular importance to this work is the fact that the automation of decision making in such systems is relevant in the era of industrial cybernetization.

The author gives a description of the Markov decision-making process and gives arguments about the advantages of POMDP over MDP.

The author's conclusions are logical and consistent, but a number of issues are not covered in the article. Notes:

  1. The choice of a decision-making method is not fully substantiated in the work, and the applications of the latter are not disclosed to the reader, given that today this specification is used in many areas, including robotics, automated control, economics and production.
  2. The author does not explain the choice of the value of the discount factor: in the classical formulation of the problem, it cannot exceed the value of unity, however, the paper presents experimental results at a higher value of this.
  3. An abstract synthetic example is not a very good way to demonstrate the proposed method. It is better to take an example with real data, especially since analogies with production processes are given.
  4. The work lacks visibility. In particular, when considering the impact of noise on the process and learning outcomes. However, analytical representations overly describe the logic of Markov processes within the framework of the considered example.
  5. Technical errors:

5.1. Line 110 "…environment tn is…"

5.2 Technical errors “Section 3 the missing data mechanisms…” “…Section 5 introduce…”.

5.3. Why are "Controller" and a number of other words capitalized? (section 3)

Reviewer 2 Report

Dear authors,

I have finished the review of your paper. It seems to offer a valuable contribution to the state of the art, however in my opinion, you need to provide more details, perhaps in a table, of the dataset employed to training and make more experiments with your proposal against other models of the state-of-art.

Reviewer 3 Report

In abstract, instead of "we presents", "we present" should be used. A space should be used before the bracket, there are a lot of examples where there is no space in the text. It is not clear enough how the terms "system state" and "system status" differ, these terms are applied consistently in such a context that they are different concepts.

A comma is required after "As an example". Instead of "optmilaity" you need to use "optimality". A space is required after the dot. In the combination “we are in interested”, the preposition “in” is not required, just “we are interested” is enough.

If Q2 is a question (lines 100-103), then this paragraph should be formulated as an interrogative sentence ending with a question mark, as in paragraph Q1 (lines 95-99).

Usually, in articles, the subject of research and the tasks solved by the article are first reported. The introduction of the article talks about the medical application for determining the dosages of patients' drugs. From this we can conclude that the article has a medical focus, that is, it is devoted to methods for solving the outlined problem. If the article is devoted to the study of methods for solving such problems as a whole, abstractly, without reference to a medical problem, then in the introduction one should start talking not about a medical problem, but in general about a class of problems that are solved by certain methods, algorithms, then the bibliography should be given precisely about the essence and level of development of these methods. In this context, the discussion of the medical problem is redundant, and the description of the problem that must be solved in order to improve the method is clearly insufficient.

A comma is required after "By designing the experiments".

One should start with the fact that there are methods for solving some similar problems, then, referring to sources, indicate what advantages and disadvantages these methods have, what problem needs to be solved to improve at least one of these methods. Next, we need a statement of the problem: what is given, what needs to be solved. The following are methods for solving this problem. This is the usual and standard way of presenting the results of solving any problems in technical and mathematical articles. It is undesirable to deviate from this way of presenting new material, since this way, IMRAD, is recognized by the scientific community as the best way to present new results in scientific articles and reports.

It is not quite correct to use the term "continuous-valued noise" (lines 175 - 176). There is no constant noise value, if the noise is constant in magnitude, then it is not a random noise process, but a deterministic noise. For stationary noise, the term "noise with constant energy" is used, or rather "noise with constant dispersion", or simply "stationary noise".

On line 181, "However, It is possible" should not use a capital letter in the word "it".

On line 202, after the words "Example 1:", the sentence begins with a small letter. A similar situation in the line after "Example 2:" 210 sentence begins with a capital letter. It is recommended that after the words "Example 1" and "Example 2" do not put a colon, but a point, and then start a new sentence with a capital letter in both cases. The ellipsis in Example 1 is unclear, see line 203. It would be better to use the phrase about some illness. Also, the use of the ellipsis in Example 2 is not clear.

Instead of "an related", "a related" should be used (line 210).

Example 1 is too hypothetical, it is clear that the authors of the article do not really understand the problem. The doctor will never adapt the dose of the drug depending on the patient's condition, the doctor's job is to make a diagnosis, clarify the tolerance of various drugs, after which certain drugs are prescribed according to the standard scheme, which depends on many factors, such as the age and weight of the patient and others. The doctor does not set up long-term experiments on each patient to adapt different doses, treatment is not scientific research to identify the patient's mathematical model as a non-stationary noisy model. This example is unsuccessful. The authors give this example twice: firstly, in the introduction, and secondly, in section 4. If this problem was actually solved by this method, then this should be taken out not as a hypothetical example, but in the “Problem Statement” section. Generally speaking, the problem of identifying an object under noisy conditions does not require confirmation of its scientific character and relevance by such invented examples in cases where the authors have nothing to do with solving problems with such examples. If the authors are working in this direction, then in the formulation of the problem it was necessary to indicate the features of this problem, the features of random factors of drug tolerance (analogues of noise), describe the results already achieved earlier, on the basis of which the formulation of the problem was carried out, then the formulation of the problem should have followed precisely in terms of medical use. The use of such a specific example as Example 1 requires, as a logical conclusion of the article in the “Experiments results discussion” section, to give an example from this area, taken in the framework of cooperation with some specialists from the field of medicine, then apply the proposed method, get some new experimental result and then discuss it. If there is no such application in the results section, then it is desirable to remove Example 1 from both the introduction and the "Problem formulation" section.

In example 2, the setting is extremely general and extremely vague; this is not a specific example, but the most general declarations. The example is not described in such non-specific terms. The following sentence is especially incomprehensible: "In these systems the main goal of the controller is to maintain the system on PREDEFINED state vectors which are usually PREDEFINED". 

The entire fourth section, starting with line 217 and ending with line 410, is supposedly a presentation of already known information from the theory.

Also presumably Section 5 is also not original, since it outlines known methods for solving the problems discussed.

Further, in Figure 7 and Table 2, some numerical model is given with specific numerical data. It is not clear where these numerical values ​​come from. Lines 560-561 say that an empirical model is being introduced. But after all, the article is devoted to such a study in which the object model is not known, and the task is to obtain an object model. It is not clear how this is related to the demonstration, where the authors know the object model in advance. Section 6.2 does not talk about noises from line 559 to line 607, but at the end of this section there are graphs that, apparently, should convince the reader of the effectiveness of the proposed method. But the task is incomprehensible: there is no noise, the model is known, what, in fact, was modeled? In the formulation of the problem, the model is not known, noise is present, then what is the connection between the topic of the article and Section 6.2 is not clear.

Figure 7 shows that Component #2 is no different from component #1, and component #5 is no different from component #3. In this case, there is no need to draw all these components, it is enough to indicate in the text that they are identical.

Table 2 actually gives the same information that the images in the form of graphs in Figure 7, apparently, it is not required to present the same information in two different ways, completely identical in content, if this text is not a textbook, but a scientific article. It is enough to indicate the correspondence of the description once in the form of a table using a single example, for example, for component #1.

In figure 8, it seems that instead of "Utilized methodology" it would be better to use the words "Results of the use of the proposed method" or more briefly: "the proposed method". It was not possible to understand where the value of the "Lambda" parameter, equal to 3.16, came from, and what will be the result if you use some other value for this parameter, except for the specified one and other than zero.

The very essence of the noise experiment seems to be contained in Section 6.4, but this section is extremely brief and of little substance.

Section 1.4 poses two questions, Q1, Q2 (lines 96-102). At the end of the article, line 609 says that answers are given to some research questions, but in fact, the questions that were formulated at the beginning of the article were not answered at the end of the article.

The advantages of the proposed method have not been demonstrated in any real-life example, so it was all the more unnecessary to talk about two practical examples at the beginning of the article. And if we are already talking about practical problems, then the solution of some practical problem with a detailed description of what, in fact, it consisted of, had to be given at the end of the article.

Sections 6.5 and 7 do not add anything specific, do not allow you to understand what the results are. The “Conclusion” section should clearly explain what problem was solved, with what result, and how a positive result is confirmed, this is not in the article.

The subject of the article may correspond to the direction of the journal "Sensors", given on the page https://www.mdpi.com/journal/sensors/sections/Intelligent_Sensors , if there is a correspondence to the section "Deep learning, big data rocessing and machine learning in sensor systems" (including blockchain, security and cyber security algorithms and architectures)”. The remaining sections are very far from the subject of the article. Then one should pay attention to the learning algorithms and demonstrate its performance on real models described in such terms so that the correctness of the simulation can be verified. In science, this criterion is decisive. See, for example, https://en.wikipedia.org/wiki/Falsifiability 

Round 2

Reviewer 2 Report

Dear authors,

I have finished the review of the revised version of your paper. I am aware of the short time to make your revisions and in my opinion you have addressed one of my greatest concerns: the hyperparameters values.

Reviewer 3 Report

The paper can be accepted in present form